# Iron and Physical Activity: Bioavailability Enhancers, Properties of Black Pepper (Bioperine^®^) and Potential Applications

**DOI:** 10.3390/nu12061886

**Published:** 2020-06-24

**Authors:** Diego Fernández-Lázaro, Juan Mielgo-Ayuso, Alfredo Córdova Martínez, Jesús Seco-Calvo

**Affiliations:** 1Department of Cellular Biology, Histology and PHArmacology, Faculty of Health Sciences, University of Valladolid, Campus de Soria, 42003 Soria, Spain; 2Department of Biochemistry, Molecular Biology and Physiology, Faculty of Health Sciences, University of Valladolid, Campus de Soria, 42003 Soria, Spain; juanfrancisco.mielgo@uva.es (J.M.-A.); a.cordova@bio.uva.es (A.C.M.); 3Institute of Biomedicine (IBIOMED), Physiotherapy Department, University of Leon, Visiting Researcher of Basque Country University, Campus de Vegazana, 24071 Leon, Spain; dr.seco.jesus@gmail.com

**Keywords:** iron, bioavailability, black pepper, piperine, supplementation, physical activity

## Abstract

Black pepper (*Piper nigrum* L.) has been employed in medicine (epilepsy, headaches, and diabetes), where its effects are mainly attributed to a nitrogen alkaloid called piperidine (1-(1-[1,3-benzodioxol-5-yl]-1-oxo-2,4 pentenyl) piperidine). Piperine co-administered with vitamins and minerals has improved its absorption. Therefore, this study aimed to describe the impact of the joint administration of iron (Fe) plus black pepper in physically active healthy individuals. Fe is a micronutrient that aids athletic performance by influencing the physiological functions involved in endurance sports by improving the transport, storage, and utilization of oxygen. Consequently, athletes have risk factors for Fe depletion, Fe deficiency, and eventually, anemia, mainly from mechanical hemolysis, gastrointestinal disturbances, and loss of Fe through excessive sweating. Declines in Fe stores have been reported to negatively alter physical capacities such as aerobic capacity, strength, and skeletal muscle recovery in elite athletes. Thus, there is a need to maintain Fe storage, even if Fe intake meets the recommended daily allowance (RDA), and Fe supplementation may be justified in physically active individuals, in states of Fe deficiency, with or without anemia. Females, in particular, should monitor their Fe hematological profile. The recommended oral Fe supplements are ferrous or ferric salts, sulfate, fumarate, and gluconate. These preparations constitute the first line of treatment; however, the high doses administered have gastrointestinal side effects that reduce tolerance and adherence to treatment. Thus, a strategy to counteract these adverse effects is to improve the bioavailability of Fe. Therefore, piperine may benefit the absorption of Fe through its bioavailability enhancement properties. Three research studies of Fe associated with black pepper have reported improvements in parameters related to the metabolism of Fe, without adverse effects. Although more research is needed, this could represent an advance in oral Fe supplementation for physically active individuals.

## 1. Introduction

Iron (Fe) is the most abundant trace mineral involved in cell metabolism and the growth of organisms. A smaller fraction (2%) localized in some proteins containing heme and Fe is present as Fe-sulfur (S) groups that contribute to physiological systems such as oxygen (O_2_) transport, DNA synthesis, metabolic energy, cellular respiration and electron transport in mitochondria. Approximately 30% and 10% of body Fe is stored as ferritin (Ft) and hemosiderin in the liver, bone marrow, and muscle. In addition, Fe can be used in erythropoiesis according to the demands of the body [1].

Physical activity (PHA) causes alterations in different cellular metabolic processes, which are reflected in modifications in biochemical and/or hematological parameters. Thus, the concentration and availability of Fe in the body is an essential factor that conditions aerobic capacity in PHA healthy individuals. Fe is a crucial mineral involved in the different physiological mechanisms related to physical performance and resistance, particularly in endurance sports. Therefore, Fe is involved in the process of oxidative phosphorylation and in the activity of antioxidant enzymes involved in muscle strength and endurance. In addition, Fe is a component of hemoglobin (Hb), which is a marker of blood O_2_ capacity linked to the sports results. Myoglobin (Mb), with a Fe atom in its structure, ensures O_2_ supply during intensive physical exercise, and exhibits antioxidant properties [2].

Thus, Fe depletion, with or without anemia, may impair aerobic physical performance. A decrease in maximal oxygen uptake during PHA has been found as a consequence of iron deficiency (ID). Indeed, marginal ID provokes a reduction in endurance capacity, only when an impoverishment of tissue and a drop in serum Ft concentration have occurred. Furthermore, Fe homeostasis is altered in some athletes competing in and training for endurance sports. In this sense, it has shown that decreased Fe storage increases fatigue, delays skeletal muscle recovery, and decreases strength and aerobic capacity. In addition, serum Fe and transferrin saturation are changeable depending on the degree of fatigue and exercise training load [3,4].

Clinical signs of ID are the most frequent cause of triggering *true anemia* in athletes [3]. However, *sports anemia*, which was first reported by Yoshimura et al. [5] is due to lower Hb concentrations than the sedentary population. This *sports anemia* is the result of a physiological response to aerobic physical exercise caused by a volume of expanded plasma that dilutes erythrocytes. *True anemia* in athletes limits sports performance, derived from prolonged strenuous exercise that directly affects Fe metabolism and reduced Hb and Ft [3]. In addition, several etiological factors may explain storage Fe depletion in PHA healthy individuals such as gastrointestinal blood loss, increased loss of Fe in sweat and urine, intestinal Fe malabsorption and malnutrition, mechanical hemolysis, and menstruation is an added risk factor for ID [6]. Furthermore, a mechanism of physical activity-induced ID involving the influence of inflammation associated with the development of PHA on the post-exercise hepcidin (HAMP) response has been proposed. The impact of PHA on HAMP-dependent control of Fe status by cytokine-induced HAMP up-regulation represents a mechanism behind ID (possibly leading to anemia) in PHA active, healthy individuals and may have important practical implications for physical performance [7].

For this reason, it is essential to maintain reserves of Fe to maintain sports performance. Therefore, supplementation may be justified in PHA active, healthy individuals, even when the diet meets the recommended daily amount for Fe [4,8,9]. Furthermore, supplementation with Fe is the most commonly used strategy to achieve adequate levels of Ft and/or avoid ID in PHA active, healthy individuals [10].

These facts may make sports nutrition researchers consider new strategies for Fe supplementation. Among these ergo-nutritional strategies is the simultaneous administration of foods whose active ingredients stimulate absorption, increasing bioavailability, and reducing the dose and, therefore, the adverse effects of Fe supplementation [11]. In this context, black pepper may be useful because piperine (alkaloid present in black pepper) co-administered with vitamins and minerals has been shown to improve their absorption. Therefore, this manuscript aims to describe the impact of co-administration of Fe plus black pepper, as a commercial preparation Bioperine^®^, on physically active healthy individuals.

## 2. Dietary Iron

Adequate nutritional intake of Fe has been established as the first control measure in ID. The absorption of heme Fe should be considered higher than that of free Fe. The two states of free Fe also have differences in intake, with ferrous ion (Fe^2+^) more widely absorbed than ferric ion (Fe^3+^) [12]. In the diet, the bioavailability of Fe varies between 5–15%, depending on the body’s reserves of Fe. In ID, the bioavailability of Fe increases to 35%. In addition, Table 1 shows that the absorption of Fe in the intestinal tract conditioned by different nutritional enhancers and inhibitors [13].

Although the recommended dietary allowance (RDA) of Fe is 15 mg/day for women and 10 mg/day for men, the estimated average requirement (EAR) should be increased by 30–70% because of the etiological factors that we have reported on in another section of this manuscript concerning athletes [10]. A diet of approximately 2500 kcal is appropriate for athletes who provide 2.3 mg Fe/day. In addition, these diets include the energy intake needed for athletes who most often suffer from ID, which may be related to their low body mass index [14,15].

## 3. Benefits of Oral Iron Supplementation

The pharmaceutical oral forms of Fe preparations are non-heme (iron salts) and heme (higher bioavailability). Oral supplements with ferrous salts (sulfate, fumarate, and gluconate) are better absorbed (10–15% bioavailability) than ferric salts of Fe complexes that are administered together with amino acids, polysaccharides, and proteins such as ovalbumin. However, high adverse gastrointestinal effects such as constipation, nausea, vomiting, diarrhea, and dark stools have been reported for ferrous Fe salts [16,17].

The amount of elemental Fe that is available to be absorbed into the body varies depending on the iron salt. Thus, the amount of elemental Fe is 20% for ferrous sulfate (FeSO_4_), 33% for ferrous fumarate, and 12% for ferrous gluconate. We must consider that if 100 mg of FeSO_4_ is equivalent to 20 mg of elemental Fe, this is approximately the RDA (18 mg/day).

It has described that 50% of patients supplemented with oral Fe have gastrointestinal adverse effects induced by the direct toxicity of ionic Fe due to its caustic action on the intestinal mucosa [18]. It should also have considered that the excess of Fe, due to the abuse of supplements, makes it difficult for the organism to get rid of Fe micronutrients caused by Ft saturation. The excess of Fe is stored in the form of hemosiderin, which causes hemosiderosis (deposits of Fe in the liver or spleen) and hemochromatosis (deposits of Fe in body tissues) [1]. As a consequence of these adverse reactions, there is a reduction in tolerance and adherence to the traditional treatment with oral Fe salts [19].

Furthermore, heme Fe supplements have a lower incidence than non-heme Fe, which could be through different routes in the absorption mechanism. Non-heme Fe by the action of stomach hydrochloric acid passes into its reduced form, Fe^2+^, which is the soluble chemical form capable of passing through the membrane of the intestinal mucosa. Although Fe can be absorbed throughout the entire intestine, its absorption is most efficient in the duodenum and the upper part of the jejunum. The intestinal mucous membrane has the facility to trap the Fe and allow its passage inside the cell, due to the existence of a specific receptor in the layer of the edge in the brush. The apotransferrin of the cytosol contributes to increasing the speed and efficiency of Fe absorption. This type of heme Fe crosses the cell membrane as an intact metalloporphyrin, once the endo-luminal or enterocyte membrane proteases hydrolyze the globin. The products of this degradation are essential for the maintenance of heme in a soluble state, thereby ensuring its availability for absorption [10].

Generally, 100–200 mg taken once or twice a day is prescribed, which is a high dose of elemental Fe [20]. The adverse reactions can be minor when a lower, more frequent, dose is dispensed. In this sense, one possibility is to use 100 mg of FeSO_4_ taken as 50 mg twice per day [19]. A daily dose of between 50 mg/day and 150 mg/day 1 may be adequate for non-anemic athletes. However, Villanueva et al. [19] used very high doses of up to 300 mg/day in bleeding patients with severe ID anemia. We have reported [4] that oral supplementation with 325 mg/day of ferrous sulfate as 105 mg/day of elemental Fe for 11 weeks does not modify Fe reserves and increases the strength of elite volleyball players. In addition, Cordova et al. [16] prevented the decrease of serum Fe, Ft, Hb and hematocrit, and improved recovery in elite cyclists during the *Vuelta a España* competition by administering 800 mg Fe protein succinate, which contained 80 mg/day Fe. In this sense, Nielsen et al. [21] showed that a fasting dose of 100 mg/day Fe ferrous administered for 3 months or more is suitable for non-anemic athletes with low serum Ft or ID only and also in anemic athletes.

An effective treatment for ID is supplementation with a dose of 60–80 mg/day of elemental Fe for 12 weeks, in healthy persons [22]. In this sense, the use of 80 mg/day of Fe achieved a decrease in fatigue in non-anemic menstruating women with low Ft [23]. However, another author [24] recommended an intake of heme Fe including meat, fish, legumes, and vegetables five times a week for healthy non-anaemic people with ID. Mielgo et al. [25] reported that an intake of 25.8 mg/day of dietary Fe is not sufficient to prevent 30% of female volleyball players from suffering from pre-latent Fe deficit and 20% from latent (pre-anemia) deficit. However, this does not guarantee high levels of Fe. Therefore, these dietary recommendations were supplemented with 28 to 50 mg of elemental Fe daily with mild gastrointestinal adverse effects [24].

## 4. Bioavailability of Dietary Iron

One strategy to counteract these adverse effects derived from supplementation and combat Fe ID is to improve the bioavailability of Fe. Bioavailability, defined as the efficiency with which Fe is obtained from the diet, is biologically used, and considered when it is necessary to enhance dietary intake [26].

Some factors modulate the bioavailability of dietary Fe, such as the content of Fe-hemic and Fe-non-hemic in foods; the original state of Fe, considering that, in an oxidized state and at a pH > 4.0, it is exceedingly insoluble and scarcely absorbed; the metabolic demand of Fe; the substances present in the diet that can be promoters that stimulate the absorption (nutritional enhancers) or that block the absorption of Fe (dietary inhibitors); the interactions between Fe and the components of the digestive tract [27]; the genetic endowment of the individual associated with the absorption of Fe, which determines the levels of Fe in serum and is determined by genes such as HFE, TMPRSS6, TFR2 and TF [28].

The bioavailability of Fe is also conditioned by some physiological factors such as body reserves of Fe, the speed of erythropoiesis, hypoxia, infections, and the mobilization of Fe and HAMP, which is a hepatic peptide intimately related to the homeostasis of this micronutrient [1,13]. Thus, there are interactions between exercise-induced factors and the expression of HAMP in humans, which are directly related to an increase or decrease in the absorption and bioavailability of Fe. Factors that negatively regulate exercise-related HAMP levels include anemia, hypoxia that triggers erythropoietin (EPO) secretion, and hemolysis [29]. On the other hand, inflammation, oxidative stress and interleukin-6 (IL-6) act by positively regulating the levels of HAMP [30]. These factors are induced during exercise, causing an alarming increase in HAMP, mainly as an acute-phase inflammatory response. Therefore, the origin of a negative iron balance in athletes may be due to factors related to exercise (type, duration, and intensity) and HAMP status. All of these directly affect iron absorption and hematological parameters. HAMP expression decreases metal absorption, increases cytoplasmic concentrations of Fe-saturated Ft, and decreases transport to the blood vessels, whereby Fe accumulates in the enterocyte and is excreted within the desquamated epithelial cells from the intestine. On the other hand, unlike excesses of Fe, deficiencies of Fe, hypoxia, and states of increased erythropoiesis generate a decrease in the expression of HAMP, and therefore, these situations are directly related to an increase in the absorption and bioavailability of Fe [13,31].

Some studies [32,33,34,35] indicate that the concomitant use of vitamin C (ascorbic acid), vitamin A, vitamin E, vitamin B9 (folic acid), vitamin B12 (cobalamin) and vitamin D3 (cholecalciferol) increases the bioavailability and absorption of Fe in the intestinal tract. These absorption enhancers can be provided by diet or direct supplementation preparations. These compounds could lead to formulations of multi-components associated with Fe that would facilitate its absorption in the digestive tract [36].

In this way, vitamin C is a promoter of the absorption of non-heme Fe. However, it does not affect heme Fe. Its effects are related to its reducing power, which blocks the synthesis of insoluble ferric hydroxide, and also its the capacity to synthesize soluble complexes with ferric ions, which maintains solubility at an alkaline pH in the duodenum. Similarly, other promoters, such as vitamins A and E, act in a similar pathway [37].

Mielgo-Ayuso et al. [35] described a positive relationship between 25(OH)D levels and levels of faith in the body in a group of elite athletes. This study reports that vitamin D3 stimulates erythropoiesis, probably because reticulocytes express receptors for the active form of vitamin D3 that would accelerate their proliferation and maturation into erythrocytes. Thus, supplementation with vitamin D3 would play a vital role in the prevention of ID and/or anemia.

The *naive* erythrocytes formed recently by erythropoiesis allow *senior* erythrocytes to be replaced daily by phagocytosis. Koury et at. [38] described the critical role that Fe, folate, and vitamin B12 play in erythropoiesis. Folate and vitamin B12 stimulate erythrocyte ontogeny from the erythroblast stage. Thus, folate and vitamin B12 deficiency inhibit purine and thymidylate synthesis, impair DNA synthesis and cause erythroblast apoptosis, resulting in ineffective erythropoiesis. The consequences of which produce anemia.

It is necessary to consider in the supplementation that the pharmaceutical form of Fe administered has a decisive influence on bioavailability. Biopharmaceutical bioavailability includes the availability, absorption, retention, and use of Fe, and is the critical factor for Fe to be biologically productive [39]. Fe preparations with higher bioavailability, which allow one to take lower absolute doses of Fe, may cause less gastrointestinal discomfort and be better tolerated by patients [21,39]. One way to increase the bioavailability of Fe is through the use of absorption enhancers, which have significant repercussions in terms of the pharmacokinetics and pharmacodynamics of Fe supplements, improving the processes of release, absorption, metabolism, utilization, and excretion [25].

## 5. Characteristics of Black Pepper as a Bioavailability Enhancer

Black pepper (*Piper nigrum* L.), is used in traditional medicine because it contains the active pharmaceutical ingredient 1-(1-[1,3-benzodioxol-5-yl]-1-oxo-2,4 pentenyl) piperidine, which is a pungent nitrogen alkaloid. Black pepper is used to treat epilepsy, headaches, and diabetes. In addition, *Piper nigrum* L. is used in the cooking of some foods with a unique flavor and as a preservative in food additives or perfumes [40].

Piperine has a low risk of toxicity, is not genotoxic and does not present any significant adverse effects in the murine model at supraphysiological doses (5–20 times higher than the average human intake) on internal organs, weight, Hb levels, total serum proteins, albumin, cholesterol, fats, and nitrogen. Its various medical properties and advantageous safety profiles facilitate the pharmaceutical employment of piperine as a dietary and health supplement. As a result, BioPerine^®^ capsules (Sabinsa Corporation, East Windsor, NJ, USA), which are a black pepper extract commercially available as a dietary supplement, are formulated up to 15 ppm as a food ingredient [40,41].

Bioperine^®^ has been administered together with vitamins, minerals, and some nutrients (Table 2; extracted from Majeed M et al. [42]) as a bioavailability enhancer [42]. The heat effect is a manifestation of the biological activity of the piperine contained in the black pepper. Bioperine^®^, a standardized extract of *Piper nigrum* L. containing 95% piperine in the form of 1-piperopiperidine, stimulates thermogenic action in the epithelial cells of the small intestine. This acts as a thermonutrient that allows for increased absorption and bioavailability of the nutrients [42,43,44].

The mechanism of action (Figure 1) responsible for the bioavailability enhancing effect of Bioperine^®^ is because a small amount of piperine stimulates the release of catecholamines, thermogenic hormones whose activity is made possible by the presence of cyclic adenosine monophosphate (cAMP) [42,43]. The thermogenic effect has been localized in the epithelial cells of the intestine that increase the absorption of nutrients. However, the catecholamine-mediated thermogenic response is relatively short. Therefore, the window of absorption has a narrow temporal scope. These thermogenic properties may explain how a minimal amount of BioPerine^®^ can produce such a powerful effect on the uptake of serum nutrients [45]. The possible justification for this mechanism would be due to modifications in the concentration of cholesterol (the central element of the plasma membrane) in the cell membranes, altering the fluidity of the lipid bilayer, and some functions of the layer such as some enzymatic activities and the processes of ion transport. In a previous study, piperine significantly lowered cholesterol concentrations in the villi. However, it did not affect the phospholipid content of the intestinal villi, resulting in a relatively lower cholesterol: phospholipid ratio in the villi of the jejunum and ileum. Those changes in the villi lipid profile might be linked to increasing the lipid fluidity of the membranes of the jejunum and ileum. These alterations of membrane fluidity directly influence the passive permeability properties of layers [46].

In another study, black pepper and piperine aroused the activities of brush border membrane enzymes, including glycyl-glycine dipeptidase, leucine aminopeptidase, g-glutamyl transpeptidase and alkaline phosphatase in the jejunal mucosa, which stimulate the active transport of amino acids. There are other pathways by which piperine stimulates nutrient absorption, including increased micelle formation and modification of the epithelial cell membrane due to piperine’s affinity to fatty substances [43]. This alteration in the dynamics of membrane lipids, together with the change in the conformation of enzymes in the intestine, would also facilitate the absorption of Fe [44]. Taken together, these results may suggest that piperine alters membrane structure by taking advantage of the apolar nature, allowing it to interact with surrounding lipids and hydrophobic portions of proteins. Perhaps all these interactions will overcome the steric hindrance of membrane lipids to proteins and thus modulate enzyme formation.

Furthermore, ultrastructural analyses with electronic scanning microscopy revealed an increase in microvilli length in the presence of piperine. It was suggested that piperine improves the permeability of the epithelial barrier of intestinal cells by altering their plasma membrane. In addition, piperine can induce the synthesis of cytoskeleton-related proteins that cause an increase in the absorption surface of the small intestine [43,44].

## 6. Effectiveness of BioPerine^®^ as a Thermonutrient (Stimulating the Bioenergetic Processes)

BioPerine^®^ with thermonutrient activity and bioavailability enhancement properties is a natural product. A thermonutrient is a nutrient with properties stimulated the bioenergetic processes in the gastrointestinal epithelium [47]. It has been observed that the administration of a preparation of beta-carotene (15 mg) plus Bioperine^®^ (5mg) increases the levels of beta-carotene by approximately 2 times in humans. Coenzyme Q10, L-(+)-selenomethionine, vitamin B6, vitamin C, and plant-based compounds, such as curcumin extract, showed improved bioavailability when given in partnership with BioPerine^®^. The administration of Bioperine (5 mg) to the preparation of vitamin C plus isopropanol only increased the bioavailability of vitamin C, but the bioavailability of alcohol was not altered [43].

With respect to Fe, Majeed et al. [45] evaluated the bioavailability of elemental Fe in combination with and without BioPerine^®^ in an animal model, with the objective of validating the capacity of BioPerine^®^ in improving the bioavailability of oral Fe. The animals were stable and followed the same diet, based on feed, provided by the animal. They found that the Fe serum concentration was significantly higher in rabbits in the Fe plus BioPerine^®^ group (36.55 ± 9.97 μg/mL at 8 h) compared to the control group, Fe without BioPerine^®^ (4.730 ± 0.94 μg/mL at 24 h); thus confirming that BioPerine^®^ could be employed as a critical natural product to increase the bioavailability of elemental Fe.

Majeed et al. [44] performed a study on regular physical activity practitioners with ID and anemia. Fe supplementation with BioPerine^®^ showed significant efficacy for 56 days, evaluated by increased Hb levels (*p* ≤ 0.0001), and it also decreased the erythrocyte sedimentation rate (*p* ≤ 0.001). An increase in serum Fe may be correlated with the immediate release of Fe from the BioPerine^®^ study supplement into the systemic circulation, thereby increasing the bioavailability of Fe over a substantial space of time. In addition, these authors described the increased health status of subjects in response to the SF-36 Health Questionnaire (*p* ≤ 0.0001) and the decreased fatigue (*p* ≤ 0.0001) assessed by the Fatigue Severity Scale during the study. A significant increase (*p* < 0.05) was observed in the Fe group with BioPerine^®^ in the levels of Hb, Ft, and total Fe fixation capacity concerning the group-administered Fe. There were no significant differences (*p* > 0.05) in the diet between both groups, evaluated by a nutritionist using the Food Frequency Questionnaire (FFQ). Therefore, the administration of Fe and BioPerine^®^ is potentially useful for the management of ID with anemia [44].

Recently in professional rowing athletes, Fernández-Lázaro et al. [36] evaluated the comparative efficacy of two oral Fe supplements, one supplement with a dose of 325 mg in the form of ferrous sulfate and another with a dose of 50 mg in the presence of BioPerine^®^ during the 10-week sports season. The athletes had more than 3 years of experience in the elite rowing team of the Rowing Club Association and took part in this randomized, non-placebo-controlled trial. The professional rowers followed a balanced diet supervised continuously by the club dietitian/nutritionist and all completed an analogous training program managed by the same graduate in Physical Education. Both supplements showed significant improvement in Ft compared to the control group, but not in sports performance. Thus, the possibility of obtaining Ft improvements through the use of six-times lower doses, together with bioavailability enhancers, could improve the risk/benefit ratio in the supplementation of athletes.

It should be noted that in the two human studies [36,44], none of the recruited subjects reported adverse events resulting from the use of Fe with BioPerine^®^, which indicates the safety and effectiveness of the supplement. However, in the future, more research is needed to re-evaluate the effectiveness of Fe and BioPerine^®^ co-administration on the hematological profile and to determine possible improvements in sports performance over more extended periods of supplementation in different sports modalities.

## 7. Safety Verification for Black Pepper Intake

Standard daily intake levels, and even 250-times higher levels, of either black pepper or its active ingredient piperine showed no adverse effects on the parameters evaluated, such as growth, organ weight, and blood components [48]. The same was found in the murine model when rats were given doses of 5 to 20 times the healthy human intake. Other parameters were also evaluated, such as total serum protein, albumin, globulin, glucose, and cholesterol, activities of serum aminotransferases and phosphatases, fat and nitrogen balance, all of which were unaltered [49]. Concerning the possible genotoxicity of piperine, all tests performed showed the non-genotoxic nature of piperine [50]. Furthermore, none of the doses of piperine administered for five consecutive days to test the immunotoxic effect demonstrated any toxicity on the immune system [51]. For these reasons, we believe the black pepper or piperine, had no apparent toxic effect and does not appear to be a poisonous chemical.

The black pepper or its active ingredient, piperine, has been reported in this manuscript to have the potential to increase the bioavailability of some nutrients, especially iron ore, andhas low toxicity. However, its potential as a potent inhibitor of cytochrome P450 and, therefore, of phase I reactions, mainly aromatic hydroxylation, should be considered when administering it [52]. As a result of the stimulation of these metabolization reactions, piperine improves the bioavailability of drugs that are metabolized in the liver; it increases their plasma half-life, delays their excretion, and increases their therapeutic potential.

## 8. Potential Application and Future Prospects of Black Pepper as an Oral Iron Supplement

Piperine has the potential to be applied in the formulation of supplements to improve sustained and controlled release bioavailability, to allow a decrease in dosage and dosing frequency, which would increase patient compliance and tolerance [53]. According to what we have described in this manuscript, it could be hypothesized that the thermogenic activity of small amounts of piperine increases the absorption rate of the nutrients administered together with piperine. Therefore, this property enables it to act as a thermonutrient on the epithelial cell membrane of the intestine. However, the features of the thermogenic response modulated by catecholamines are moderately ephemeral. Therefore, the time window of thermogenesis-stimulated absorption in the intestinal tract and the potential increase in bioavailability of piperine-co-administered nutrients is temporarily narrow. Therefore, the activity induced by piperine is limited in time, which makes the simultaneous administration of Fe and Bioperine^®^ necessary.

The frequent use of Fe supplements with BioPerine^®^ would stimulate the possibility of achieving more significant positive effects on some of the parameters involved in the metabolism of Fe, with lower doses, which reduces the risks of Fe overload on the gastrointestinal tract and possible deposits in organs and tissues, and attenuates the potent pro-oxidant effect of Fe. The accumulation of Fe inside the cells increases their oxidative stress, which translates into a high production of free radicals and reactive oxygen and nitrogen species, which contributes directly to muscle damage. This situation is especially critical in athletes, where the skeletal muscle has been damaged due to physical activity; one could add the pro-oxidant event of excess Fe derived from hyper-supplementation. Therefore, muscle damage will be exacerbated, which directly affects health and physical performance.

However, iron metabolism is not the only player affecting muscle performance. Due to the complexity of muscle physiology, other factors can have a decisive influence on skeletal muscle, such as age, lipotoxicity, muscle stiffness, and the role of caloric restriction in muscle health [54,55,56]. In this way, sarcopenia is the age-associated loss of muscle mass and function. Aging is a physiological process involving biochemical and morphological changes that are affected throughout the muscle [54]. These biological aging processes include reduced regenerative capacity by stem cell exhaustion, cellular senescence, increased pro-inflammatory intercellular signaling, insulin resistance, lipotoxicity, dysregulated nutrient sensing, mitochondrial dysfunction and increased oxidative stress [54,56]. This damage is associated with inflammation, protein catabolism, myocyte apoptosis, alteration of functional capacity, muscle atrophy, increase muscle stiffness, and, therefore, decreased muscle performance [56].

Moreover, histopathological muscle changes as a result of the pathways triggered by sarcopenia, which affect muscle performance, are changes in muscle fiber size and number, causing mass muscle reduction with age and selective atrophy of muscle fibers [54,56]. In addition, another factor to consider in performance and muscle function is caloric restriction (CR) [55]. CR-induced effects are related to aging in skeletal muscle. In an animal model, Chen et al. [55] observed that CR modulates mTOR signaling and the ubiquitin-proteasome pathway (UPP). CR did not alter mTOR signaling and UPP in the soleus muscles of young and adult rats, whereas CR altered the pathways in middle-aged rats, stimulating muscle protein degradation [55].

Finally, we believe that more studies are necessary because the pharmacokinetics of drugs and natural products is modulated with administered of piperine, which could enhance both therapeutic and adverse effects. For this reason, it is necessary to know the set of interactions between piperine and Fe by studying the effects of variations in dose, dose frequency and duration of treatment on the ability to modify (stimulating or inhibiting) the activity of metabolic enzymes and transporters, depending on the conditions of the treatment administered. For safe application of piperine and iron, clinical studies with long-term interventions should be conducted. This will allow the therapeutic action of piperine to be exploited in formulations with other products. Thus, more research is needed, and this may lead to an advance in oral Fe supplementation for physically active healthy individuals.

## 9. Conclusions

Supplementation with a natural product, Bioperine^®^, in the form of black pepper, as a bioavailability enhancer, could have a potential application as an athlete Fe supplement for improving hematological parameters. Therefore, it is proposed as an innovative food therapy without side effects in ID patients supplemented with Fe for the control of Fe status with or without anemia.

## Figures and Tables

**Figure 1 nutrients-12-01886-f001:**
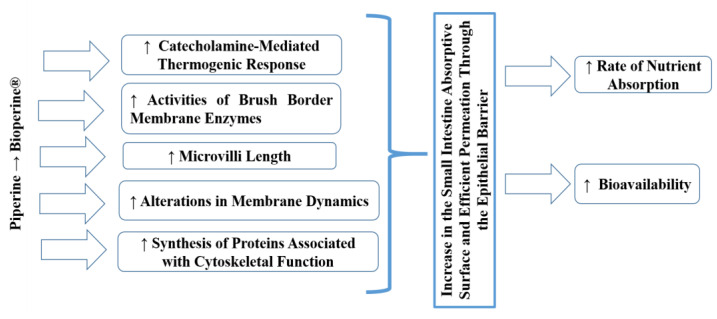
Action mechanism of piperine/Bioperine^®^ as a bioavailability enhancer.

**Table 1 nutrients-12-01886-t001:** Nutritional Enhancers and Inhibitors of Iron Absorption.

Nutritional Enhancers	Nutritional Inhibitors
Vitamin C	Phytates
Peptides from partially digested muscle tissue	Oxalates
Fermented food	Polyphenols: coffee and black tea
Organic acids: malate or citrate	Peptides from partially digested vegetable proteins
Meat	
Fish and Shellfish protein	Minerals: calcium

**Table 2 nutrients-12-01886-t002:** Nutritional elements co-administrated with BioPerine^®^ that increase absorption.

Group Elements	Nutritional Elements
**Herbal Extracts Curcuminoids**	Curcuminoids, Boswellia Serrata Extract, Ashwagandha, Ginkgo Biloba Extract, Capsaicin, Bioflavonoids.
**Water-soluble Vitamins**	B1, B2, B3, B6, B9, B12, C.
**Liposoluble Vitamins**	A, D, E, K.
**Antioxidants**	vitamin A, vitamin C, vitamin E, alPHA-carotene, beta-carotene, beta-cryptoxantine, Lycopene, lutein/zeaxanthin, pine bark bioflavonoid complex, Ge, Se, Zn.
**Amino acids**	lysine, isoleucine, leucine, threonine, valine, tryptoPHAn, phenylalanine and methionine
**Minerals**	Fe, Zn, V, Se, Cr, I, K, Mn, Cu, Ca, Mg.

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
