# Peer review of "Iron and Physical Activity: Bioavailability Enhancers, Properties of Black Pepper (Bioperine®) and Potential Applications"

_nutrients, 2020, doi:10.3390/nu12061886_

Round 1

Reviewer 1 Report

This review was very confusing to read due to the unstructured configuration.

In order to further improve the completeness of the article, it is necessary to clarify the purpose of the review at the first stage and set sub-topics suitable for the overall composition.

The purpose or goal of the study should be briefly stated for each sub-topics, and then key information (evidence for the hypothesis, etc.) should be delivered.

In particular, in order to demonstrate the expected effectiveness presented by the author, it is necessary to present evidence (references) recognized by prominent scientists or verified in many reports.

Author Response

Point-by-Point Response to Reviewer’s Comments

We would like to sincerely thank the reviewer for his/her helpful recommendations again. We have seriously considered all the comments and carefully revised the manuscript accordingly. Revisions are highlighted in pink through the manuscript to indicate where changes have taken place. We feel that the quality of the manuscript has been significantly improved with these modifications and improvements based on the reviewers’ suggestions and comments. We hope our revision will lead to an acceptance of our manuscript for publication Nutrients.

Also, the manuscript has undergone English language editing by a Professor at the Faculty of Translation, University of Valladolid. The text has been checked for correct use of grammar and common technical terms, and edited to a level suitable for reporting research in a scholarly journal.

In advance,

King regards

Rev: This review was very confusing to read due to the unstructured configuration

Authors: Thank you for your recommendation. The authors have restructured and renamed the different sections, which are reflected in the manuscript. In addition, a new section has been added at the suggestion of another reviewer

Rev: In order to further improve the completeness of the article, it is necessary to clarify the purpose of the review at the first stage and set sub-topics suitable for the overall composition.

Authors: Thank you for your observation. The authors have included a new paragraph in the introduction

These facts may make sports nutrition researchers may consider new strategies for Fe supplementation. Among these ergo-nutritional strategies could be the simultaneous administration of foods whose active ingredients stimulate absorption, increasing bioavailability, and reducing the dose and, therefore, the adverse effects of Fe supplementation (doi: 10.1016/j.freeradbiomed.2016.01.016). In this context, black pepper could be useful, because of the history that piperine (alkaloid present in black pepper) co-administered with vitamins and minerals has improved its absorption. Therefore, this manuscript aimed to describe the impact of co-administration of Fe plus black pepper, as a pharmaceutical preparation Bioperine®, on physically active healthy individuals.

Rev: The purpose or goal of the study should be briefly stated for each sub-topic, and then key information (evidence for the hypothesis, etc.) should be delivered.

Authors: Thank you for your recommendation. The authors review the impact of the joint administration of iron plus black pepper in physically active healthy individuals. Each of the sections does not serve an individual purpose but are necessary for the reader to understand in the context of physical activity and iron supplementation. In this way, the purpose of this review can be understood. The sections of this manuscript related to black pepper provide the key information that is in the scientific literature to date. The manuscript opens a way of reflection for new nutritional strategies based on the improvement of bioavailability of nutrients using bioavailability enhancers. Particularly, the iron supplementation in physical activity where the authors have some experience for the recent publication of some studies (doi: 10.3390/nu11030500, 10.3390/nu10101526).

Rev: In particular, in order to demonstrate the expected effectiveness presented by the author, it is necessary to present evidence (references) recognized by prominent scientists or verified in many reports.

Authors: Thank you for your observation. Currently, only 2 studies have used of black pepper as an iron bioavailability enhancer in physical activity. Although, as you rightly say, they are not from high impact publications. This literature review opens the research field to new studies aimed to re-evaluate the effectiveness of Fe and BioPerine® co-administration on the hematological profile to determine possible improvements in sports performance over longer periods of supplementation in different sports modalities.

Also, our studies on this topic was reflected in 2 oral communications at international congresses in 2019. In this sense, our study "Comparative efficacy of two iron oral supplements on hematological profile and sports performance in professional rowers: Evaluation of a BioPerine® Bioavailability Enhancer" was presented at the Ibero-American Congress of Nutrition held in Pamplona (Spain) in July 2019. In addition, this study was published in abstract format in Rev Esp Nutr Hum Diet. 2019, 23, 174-75.  In the same way, another study our "Impact of Bioavailability on Iron Deficiency: Nutritional effect of Bioperine® and its Potential of Application in Sports Supplementation" was presented at the International Congress of Sports and Health Sciences SPORTIS 2019, held in Pontevedra (Spain) in November 2019. Thus, you might consider the experience of the authors in that field

Reviewer 2 Report

In this work Fernández-Lázaro et al. review the impact of the joint administration of iron (Fe) plus black pepper in physically active healthy individuals. This literature review opens the research field to new studies aimed to re-evaluate the effectiveness of Fe and BioPerine® co-administration on the hematological profile to determine possible improvements in sports performance over longer periods of supplementation in different sports modalities.

The reviewer offers the following minor critiques:

In line 79 is not clear what the authors mean by “This type of anemia is caused by insufficient uptake of faith”. Please revise or clarify

The authors correctly discuss that exacerbated skeletal muscle damage can directly affect health and physical performance, however iron metabolism is not the only player. To give the readership a wider breath of the complexity of muscle physiology, can the authors give a brief introduction discussing the importance of age, lipotoxicity, muscle stiffness and the role of calorie restriction in muscle health while citing the below listed evidence? (PMID: 31227962, 31865527, 31676964)

Author Response

Point-by-Point Response to Reviewer’s Comments

We would like to sincerely thank the reviewer for his/her helpful recommendations again. We have seriously considered all the comments and carefully revised the manuscript accordingly. Revisions are highlighted in yellow through the manuscript to indicate where changes have taken place. We feel that the quality of the manuscript has been significantly improved with these modifications and improvements based on the reviewers’ suggestions and comments. We hope our revision will lead to an acceptance of our manuscript for publication Nutrients.

Also, the manuscript has undergone English language editing by a Professor at the Faculty of Translation, University of Valladolid. The text has been checked for correct use of grammar and common technical terms, and edited to a level suitable for reporting research in a scholarly journal.

In advance,

King regards

Rev: In line 79 is not clear what the authors mean by “This type of anemia is caused by insufficient uptake of faith”. Please revise or clarify

Authors: Thank you for your recommendation. The authors have corrected the error and restructured the sentence

Clinical signs of ID are the most frequent cause of triggering “true anemia” in athletes [3]. However, the “sports anemia" (wrong name) which was first reported by Yoshimura et al. [5] is due to lower Hb concentrations than the sedentary population. This “sport anemia” is the result of a physiological response to aerobic physical exercise caused by a volume of expanded plasma that dilutes red blood cells. True anemia in athletes limits sports performance, derived from prolonged strenuous exercise that directly affects Fe metabolism and reduced Hb and Ft [3].

Rev: The authors correctly discuss that exacerbated skeletal muscle damage can directly affect health and physical performance, however iron metabolism is not the only player. To give the readership a wider breath of the complexity of muscle physiology, can the authors give a brief introduction discussing the importance of age, lipotoxicity, muscle stiffness and the role of calorie restriction in muscle health while citing the below listed evidence? (PMID: 31227962, 31865527, 31676964).

Authors: However, iron metabolism is not the only player affecting muscle performance. Due to the complexity of muscle physiology, there are other factors that can have a decisive influence on skeletal muscle such as age, lipotoxicity, muscle stiffness, and the role of caloric restriction in muscle health (1, 2, 3).  In this way, sarcopenia is the loss of muscle mass and function age-associated. Aging is a physiological process involving biochemical and morphological changes that are affected throughout the muscle (1). These biological aging processes include reduced regenerative capacity by stem cell exhaustion, cellular senescence, increase pro-inflammatory intercellular signaling, insulin resistance, lipotoxicity, dysregulated nutrient sensing, mitochondrial dysfunction and increased oxidative stress (1, 2). This process damage is associated with inflammation, protein catabolism, myocyte apoptosis, alteration of functional capacity, muscle atrophy, increase muscle stiffness, and therefore decreased muscle performance (2). Moreover, histopathological muscle changes as a result of the pathways triggered by sarcopenia that affect muscle performance are changes in muscle fiber size and number by muscle mass reduction with age and selective atrophy of muscle fibers (1,2). In addition, another factor to consider in performance and muscle function is the caloric restriction (CR) (3). CR-induced effects are related to aging on skeletal muscle. In the animal model, Chen et al. (3). have observed that CR modulates on mTOR signaling and the ubiquitin-proteasome pathway (UPP). CR did not alter mTOR signaling and UPP in the soleus muscles of young and adult rats whereas CR altered the pathways in middle-aged rats stimulating muscle protein degradation (3).

References:

  1. PMID: 31227962
  2. PMID: 31865527
  3. PMDI: 31676964

Reviewer 3 Report

Overall Iron and Physical activity is a well written manuscript.

The authors mention how the use of black pepper may enhance the nutrient absorption, but perhaps the authors should also include the adverse effects of this compound. The mechanism of increasing the absorption results in non-specific and higher absorption of drugs in the system, which may not be desirable at all times.

Other than that, I have some minor suggestions:

The authors use non-canonical abbreviations for some proteins including ferritin and hepcidin which are usually referred as Ft and HAMP, respectively.

L52- Fe is present in cells as Fe-S groups and not just the S groups.

Several facts throughout the manuscript are missing the references (ex L144-L151)

There is a typo in L196..... 

Please provide ref for stating that anemia, hemolysis etc are excercise modulating factors (L197)

Author Response

Point-by-Point Response to Reviewer’s Comments

We would like to sincerely thank the reviewer for his/her helpful recommendations again. We have seriously considered all the comments and carefully revised the manuscript accordingly. Revisions are highlighted in green through the manuscript to indicate where changes have taken place. We feel that the quality of the manuscript has been significantly improved with these modifications and improvements based on the reviewers’ suggestions and comments. We hope our revision will lead to an acceptance of our manuscript for publication Nutrients.

Also, the manuscript has undergone English language editing by a Professor at the Faculty of Translation, University of Valladolid. The text has been checked for correct use of grammar and common technical terms, and edited to a level suitable for reporting research in a scholarly journal.

In advance,

King regards

Rev: The authors mention how the use of black pepper may enhance the nutrient absorption, but perhaps the authors should also include the adverse effects of this compound. The mechanism of increasing the absorption results in non-specific and higher absorption of drugs in the system, which may not be desirable at all times.

Authors (1): Thank you for your recommendation. The authors have included a new section.

Is it safe to ingest black pepper?

Standard daily intake levels and even 250 times higher of either black pepper or its active ingredient piperine showed no adverse reactions in the parameters evaluated such as growth, organ weight, and blood components (Srinivasan and Satyanarayana, 1981). The same occurs in the murine model when rats were given doses of 5 to 20 times the normal human intake. Other parameters were also evaluated, such as total serum protein, albumin, globulin, glucose, and cholesterol, activities of serum aminotransferases and phosphatases, fat and nitrogen balance, all of which were unaltered. (Bhat and Chandrasekhara, 1986). Concerning the possible genotoxicity of piperine, all tests performed showed the non-genotoxic nature of piperine. (Karekar et al., 1996). Besides, none of the doses of piperine administered for 5 consecutive days to test the immunotoxic effect, had demonstrated any toxicity on the immune system. (Dogra et al., 2004 doi: 10.1016/j.tox.2003.10.006). For these reasons, we believe by all of these evaluations of the toxicity of black pepper or piperine, had no obvious toxic effect, so piperine appears to be a toxic chemical.

The black pepper or its active ingredient, piperine, has been reported in this manuscript to have the potential activity to increase the bioavailability of some nutrients and especially iron ore and its low toxicity. However, it's potential as a potent inhibitor of cytochrome P450 and, therefore, of phase I reactions, mainly aromatic hydroxylation, should be considered when administering it (Srinivasan et at. (2007)doi: 10.1080/10408390601062054). As a result of the stimulation of these metabolization reactions those drugs that are metabolized in the liver, piperine improves their bioavailability, increases their plasma half-life, delays their excretion, and increases the therapeutic potential.

Authors (2): Thank you for your recommendation. The authors had already described a paragraph in two section on the manuscript, that could complement this new section.

  • “Bioavailability Enhancer Properties of Black Pepper”

Piperine has a low risk of toxicity, is not genotoxic and did not present any significant adverse effects in the murine model at supraphysiological doses (5-20 times higher than average human intake) on internal organs, weight, Hb levels, total serum proteins, albumin, cholesterol, fats, and nitrogen. The various medical properties and advantageous safety profiles facilitate the pharmaceutical employment of piperine as a dietary and health supplement. As a result, BioPerine® capsule (Sabinsa Corporation, East Windsor, NJ, USA) formulated, which is a black pepper extract commercially available as a dietary supplement, with safety as a food ingredient up to 15 ppm [38-39].

  • “Potential applications”

However, we believe that is necessary more studies because the pharmacokinetics of drugs and natural products administered in association with piperine is modulated, which could enhance both therapeutic and adverse effects. For this reason, it is necessary to know the set of interactions between piperine and Fe by studying various conditions of dose, dose frequency and duration of treatment by the ability to modify (stimulating or inhibiting) the activity of metabolic enzymes and transporters, depending on the conditions of the treatment administered. For the potential safe application of piperine and iron, clinical studies with long-term interventions should be conducted. This will allow the therapeutic action of piperine to be exploited in formulations with other products. Thus, more research is needed, could represent an advance in oral Fe supplementation for physically active healthy individuals.

Rev: The authors use non-canonical abbreviations for some proteins including ferritin and hepcidin which are usually referred as Ft and HAMP, respectively.

Author: Thank you for your recommendation. The authors have changed the abbreviations in the manuscript.

Rev: L52- Fe is present in cells as Fe-S groups and not just the S groups.

Authors: Thank you for your observation. The authors have corrected the error

A smaller fraction (2%) localized in some proteins containing heme and Fe is present as Fe-sulfur (S) groups is involved in biological systems such as…

Rev: Several facts throughout the manuscript are missing the references (ex L144-L151).

Authors: Thank you for your observation. The authors have included the references.

In non-athletic individuals (women and children), have been used high dosage of Fe (e.g., 60-80 mg of elemental Fe per day for 12 week) as effective treatment of (doi: 10.1503/cmaj.110950). In addition, Fe supplementation 80 mg-1 day produces a decrease in fatigue in non-anemic, non-athletic menstruating women with low FER (doi: 10.4414/smw.2017.14434)

Rev: There is a typo in L196...

Authors: Thank you for your observation. The authors have corrected the error

Thus, there are interactions between exercise-induced factors and Hep expression in humans

Rev: Please provide ref for stating that anemia, hemolysis etc are excercise modulating factors (L197)

Authors: Thank you for your observation. The authors have included the references.

Exercise modulating factors such as anemia, hypoxia which triggers the erythropoietin (EPO) and hemolysis (doi: 10.3389/fphys.2013.00332), negatively regulate Hep levels (these situations are directly related to an increase in the absorption and bioavailability of Fe) and upon inflammatory, oxidative stress and interleukin 6 (IL-6) act positively regulate Hep levels (doi: 10.1182/blood-2018-06-856500)

Round 2

Reviewer 1 Report

This manuscript has been improved in many sections, but typos are still found and minor corrections are required.

Table captions should be prepared in accordance with the Journal guideline.

References should be noted in footnote or content, not in the captions.

ex) Table 1. Nutritional Enhancers and Inhibitors of Iron Absorption [13].

Table 1. Nutritional enhancers and inhibitors of iron absorption.

ex) Table 2. Nutritional elements that increase absorption co-administration with BioPerine®.

Extracted from Majeed M et al. [42].

It is recommended to improve the subtitles as follows:

3. Oral Iron Supplementation -> Benefits of oral iron supplementation

4. Iron Bioavailability -> Bioavailability of dietary iron

5. Bioavailability Enhancer Properties of Black Pepper -> Characteristics of black pepper as a bioavailability enhancer

7. Is it safe to ingest black pepper? -> Safety verification for black pepper intake

8. future perspectives -> Potential application and future prospects of black pepper as an oral iron supplement

Author Response

Point-by-Point Response to Reviewer’s Comments

We would like to sincerely thank the reviewer for his/her helpful recommendations again. We have seriously considered all the comments and carefully revised the manuscript accordingly. Revisions are highlighted in pink through the manuscript to indicate where changes have taken place. We feel that the quality of the manuscript has been significantly improved with these modifications and improvements based on the reviewers’ suggestions and comments. We hope our revision will lead to an acceptance of our manuscript for publication Nutrients.

Also, the manuscript has undergone English language editing by MDPI. The text has been checked for correct use of grammar and common technical terms, and edited to a level suitable for reporting research in a scholarly journal.

In advance,

King regards

  • Rev: Table captions should be prepared in accordance with the Journal guideline. References should be noted in footnote or content, not in the captions.

Authors: Thank you for your observation. The authors have changed the tables captions.

  • In the diet, the bioavailability of Fe varies between 5%-15%, depending on the body's reserves of Fe. In ID, the bioavailability of Fe increased to 35%. Also, the absorption of Fe in the intestinal tract conditioned by different nutritional enhancers and inhibitors (Table 1. Nutritional Enhancers and Inhibitors of Iron Absorption [13]) [13, 14].

Table 1. Nutritional Enhancers and Inhibitors of Iron Absorption

  • Bioperine® has been administered together with vitamins, minerals, and some nutrients (Table 2. Nutritional elements that increase absorption co-administration with BioPerine®. Extracted from Majeed M et al. [42]) as a bioavailability enhancer [42].

Table 2. Nutritional elements that increase absorption co-administration with BioPerine®. 

  • Rev: It is recommended to improve the subtitles as follows:

Authors: Thank you for your recommendation. The authors have modified the titles of the sections

  1. Benefits of oral iron supplementation

  1. Bioavailability of dietary iron

  1. Characteristics of black pepper as a bioavailability enhancer

  1. Safety verification for black pepper intake

  1. Potential application and future prospects of black pepper as an oral iron supplement
